# *Bifidobacterium longum* Subsp. *infantis* Promotes IgA Level of Growing Mice in a Strain-Specific and Intestinal Niche-Dependent Manner

**DOI:** 10.3390/nu16081148

**Published:** 2024-04-12

**Authors:** Mengfan Ding, Bowen Li, Haiqin Chen, Reynolds Paul Ross, Catherine Stanton, Jianxin Zhao, Wei Chen, Bo Yang

**Affiliations:** 1State Key Laboratory of Food Science and Resources, Jiangnan University, Wuxi 214122, China; dingmf0821@163.com (M.D.); lbw1965841356@gmail.com (B.L.); haiqinchen@jiangnan.edu.cn (H.C.); zhaojianxin@jiangnan.edu.cn (J.Z.); chenwei66@jiangnan.edu.cn (W.C.); 2School of Food Science and Technology, Jiangnan University, Wuxi 214122, China; 3International Joint Research Center for Probiotics & Gut Health, Jiangnan University, Wuxi 214122, China; p.ross@ucc.ie (R.P.R.); catherine.stanton@teagasc.ie (C.S.); 4APC Microbiome Ireland, University College Cork, T12 R229 Cork, Ireland; 5Teagasc Food Research Centre, Moorepark, Fermoy, P61 C996 Cork, Ireland; 6National Engineering Research Center for Functional Food, Jiangnan University, Wuxi 214122, China

**Keywords:** *Bifidobacterium longum* subsp. *infantis*, IgA level, sIgA-coated bacteria, gut microbiota, intestinal mucosal immunity

## Abstract

Throughout infancy, IgA is crucial for maintaining gut mucosal immunity. This study aims to determine whether supplementing newborn mice with eight different strains of *Bifidobacterium longum* subsp. *infantis* might regulate their IgA levels. The strains were gavaged to BALB/C female (*n* = 8) and male (*n* = 8) dams at 1–3 weeks old. Eight strains of *B. longum* subsp. *infantis* had strain-specific effects in the regulation of intestinal mucosal barriers. B6MNI, I4MI, and I10TI can increase the colonic IgA level in females and males. I8TI can increase the colonic IgA level in males. B6MNI was also able to significantly increase the colonic sIgA level in females. B6MNI, I4MI, I8TI, and I10TI regulated colonic and Peyer’s patch IgA synthesis genes but had no significant effect on IgA synthesis pathway genes in the jejunum and ileum. Moreover, the variety of sIgA-coated bacteria in male mice was changed by I4MI, I5TI, I8TI, and B6MNI. These strains also can decrease the relative abundance of *Escherichia coli*. These results indicate that *B. longum* subsp. *infantis* can promote IgA levels but show strain specificity. Different dietary habits with different strains of *Bifidobacterium* may have varying effects on IgA levels when supplemented in early infancy.

## 1. Introduction

The antibody known as immunoglobulin A (IgA) is mostly found on the surface of mucous membranes, including those of the respiratory, urinary, and intestinal systems. It is essential for preserving mucosal integrity and preventing inflammation of the intestinal barrier [1]. Research has shown that the development of the mucosal immune system and the synthesis of sIgA are facilitated by commensal bacteria [2]. In the intestinal mucosa, intestinal microorganisms regulate both T cell-dependent and non-dependent mechanisms, which transform B cells into IgA plasma cells. This process results in the production of IgA. Thus, the higher prevalence of immune-mediated illnesses may be related to abnormalities in the baby gut microbiota [3,4]. In fact, infants at risk of asthma have been found to display “transient” gut microbial dysbiosis. [5]. In another study, microbial dysbiosis at 3 months of age was associated with the development of atopic wheeze at 5 years of age [6]. sIgA is the most abundant antibody in breast milk and functions as secreted IgA (sIgA) at the mucosal site due to its irreversible binding to secretory factors during trans-epithelial transmigration. Infants cannot synthesize IgA within the first month of life and can only obtain IgA through breast milk at a level of 0.25–0.5 g/day [7]. Research has demonstrated that a deficit in sIgA impacts the diversity of the human and mouse microbiomes, which in turn results in dysbiosis of the gut microbiota and illnesses including colitis and diarrhea. Through bacterial glycan binding or antigen–antibody interactions, sIgA can encourage bacterial colonization and proliferation in the gut [8]. sIgA deficiency increases the number and duration of colonization by Enterobacteriaceae in the gut [9]. In contrast, other bacteria show stronger colonization when coated by sIgA [10]. sIgA protects the gut microbiota by encouraging the colonization of some anaerobic bacteria and inhibiting the development of anaerobic bacteria that cause inflammation, such as Enterobacteriaceae [11].

Dietary habits are crucial in controlling IgA levels [12]. Positive dietary habits can supply probiotics with the nutrients they need, including dietary fiber and prebiotics that aid in the growth and reproduction of probiotics [13]. In addition, foods rich in probiotics in the diet, such as yogurt, fiber, and bean products, can also help increase the level of probiotics in the intestine [14]. Dietary fiber is one of the primary sources of nourishment for intestinal *Bifidobacterium*. Consuming enough dietary fiber might encourage the growth of intestinal *Bifidobacterium*, which may in turn control IgA levels [15]. *Bifidobacterium* predominates in the infant intestines while they are breastfed. The primary cause is the oligosaccharides found in breast milk, which promote *Bifidobacterium* development and reproduction. However, the relative quantity of *Bifidobacterium* was reduced in the intestines of some infants who were fed formula milk [16]. Also, compared to breastfeeding, infants fed with formula milk have an increased risk of developing diseases [17]. The intake of probiotics in the diet can to some extent alleviate the impact of formula feeding [18]. This also indicates that regardless of age, dietary habits and structure are crucial for healthy development.

Researchers have reported that probiotics, including *Bifidobacterium* and *Lactobacillus,* promote sIgA levels. For example, in in vitro experiments, *Bifidobacterium bifidum* OLB6378 increased sIgA levels, while inactivation of this strain and supplementation of full-term infants increased the amount of sIgA in the feces of full-term infants [19]. Supplementation with *B. longum* subsp. *infantis* M-63 significantly upregulated the relative abundance of *Bifidobacterium*, resulting in elevated fecal IgA levels [20]. Additionally, *B. animalis* positively impacted IgA levels in a mouse model and improved intestinal barrier function [21]. *B. animalis* BB12 relieved colic in children and increased sIgA levels [22]. Thus, certain commensal bacteria can stimulate intestinal IgA production [23,24] and improve the Th1 immune response [25]. IgA also coats the gut commensal bacteria, increasing immunological development and exposure to the immune system in early childhood according to recent research [26]. To maintain intestinal homeostasis, sIgA modulates mucosal dendritic cells with tolerogenic properties by encasing probiotic bacteria, such as *Lactobacillus rhamnosus* [27]. Consequently, sIgA-coated bacteria may be used to assess their probiotic ability. Moreover, it has been discovered that immunity is influenced by gender variations independently of microbiota. This further contributes to the variations in immunity between genders by causing the selection of a gender-specific microbiome [28]. The aforementioned study, however, primarily investigated whether different *Bifidobacterium* or *Lactobacillus* can regulate IgA, and the ability of different strains of *Bifidobacterium* or *Lactobacillus* to regulate IgA is different remains unknown. 

*Bifidobacterium* dominates in the infant intestine, especially *B. longum* subsp. *infantis* [29], which is associated with healthy infant development. As a result, it is worthwhile to investigate the *B. longum* subsp. *infantis* strains’ capacity to regulate IgA levels. We speculate that giving mice supplements of these strains may affect the IgA levels of their progeny in a strain-specific and intestinal niche-dependent way.

## 2. Materials and Methods

### 2.1. Bacteria

Eight strains of *B. longum* subsp. *infantis* were deposited at Jiangnan University, Wuxi, China. Among the eight strains, B6MNI (CCFM1269) was isolated from human breastmilk, and the other seven strains were isolated from infant feces.

### 2.2. Animal Experiments

The animal study was approved by Jiangnan University on 15 September 2021 (JN.No20210915b0601125[299]). Six-week-old male and female BALB/c mice who were specifically pathogen-free (SPF) were bought from Beijing Vital River Laboratory Animal Technology Co., Ltd. (Hangzhou, China). After a week of adaptation, male (20 ± 1 g) and female (18 ± 1 g) mice (2:1) were housed in cages at 20–26 °C and 40–70% humidity. After confirming that the female mice were pregnant, the male mice were removed. The pregnant mice were not given any medication before giving birth. Both males and females were gavaged with normal saline and *B. longum* subsp. *infantis* strains I2MI, I4MI (CCFM 1270), I4MNI, I5TI, I6TI, I8TI (CCFM1271), I10TI (CCFM1271), and B6MNI (CCFM1269), with *n* = 8 per group, from 1 week old to 3 weeks old, with 1 × 10^9^ CFU/day/mice (Figure 1). At the time of testing, groups consisting of more than eight mice following litter merging were randomly divided into eight groups. Mice were sacrificed at 3 weeks old. A blood sample (100 μL) and the jejunum, ileum, Peyer’s patch (PP), and colon were collected. The other samples were stored at −80 °C for further analysis. 

### 2.3. Biochemical Indicator Measurement

PBS (Phosphate Buffered Saline; mass: volume = 1:9) was used to homogenize 50 milligrams of the colon, which was centrifuged for 10 min at 4000× *g*. The supernatant was then collected for further analysis. The levels of IgA and sIgA in the colon were accessed with ELISA kits based on the instructions.

### 2.4. qRT-PCR Analysis

Total RNA was extracted from tissues, including the Peyer’s patches, the colon, the jejunum, and the ileum, using the TriZol method. Briefly, 10 mg of tissue was combined with 1 mL TriZol and homogenized for 30 s three times, and 200 μL trichloromethane was added and the mixture was left to stand at room temperature for 15 min. Then, the supernatant was collected after being centrifuged at 4 °C and 12,000× *g* for 15 min. The same volume of isopropanol as the supernatant was added and the mixture remained at −20 °C for 30 min. The supernatant was discarded after being centrifuged at 4 °C and 12,000× *g* for 15 min. Finally, the sediment was dissolved in DEPC water after being washed with 75% ethanol three times. RNA concentration was determined by the Nanodrop method before reverse transcription with the HiScript III All-in-one RT SuperMix. The resulting cDNA template was then kept at −80 °C. qRT-PCR was utilized for amplification, and the Ct value of each template was measured and computed using the 2^−ΔΔCt^ technique, with β-actin serving as the internal standard for relative quantification. The qRT-PCR system consisted of 95 °C for 5 min, 95 °C for 10 s, 60 °C for 30 s for 40 cycles; the melt curves depended on the mode of the machine. The primer sequences are listed in Table 1.

### 2.5. Separation of sIgA-Coated Bacteria

Ten mg of feces (wet weight) was washed with PBS (including 0.5% L-cysteine), and the bacteria pellets were collected. Bacterial pellets were blocked by goat serum and incubated by biotinylated anti-mice IgA and streptavidinized magnetic beads. sIgA-coated bacteria were collected through the adsorption of magnetic beads by magnetic poles as previously described. A suspension of bacteria was made in PBS with 0.5% L-cysteine (PBSL). After that, the suspension was incubated for 20 min with 500 µL of goat serum. The bacterial sediment was then recovered following a 5 min centrifugation at 6000× *g* and 4 °C. After the bacterial mass was suspended in PBSL, 500 µL of carboxyl magnetic beads and 20 µL of IgA antibody were each added and incubated for 20 min. Lastly, sIgA-coated bacteria from the bacterial sediment were separated using a magnetic rack washed with PBS three times for DNA extraction.

### 2.6. DNA Extraction and 16S rRNA Gene Sequencing

MP Biomedicals (Irvine, CA, USA) manufactured the FastDNA Spin Kit for Feces, which was used to extract DNA from fecal samples. The sIgA-coated bacteria isolated as specified in Section 2.5 was taken to extract bacterial DNA. Briefly, a Lysing Matrix E tube included in the MP Biomedicals kti was filled with bacterial sediment. After adding and vortexing for 10 to 15 s, 825 µL of sodium phosphate buffer and 275 µL of pre-lysis dissolving solution were added. Centrifugation at 14,000× *g* for 5 min was then performed, and the supernatant was disposed of. The mixture was then mixed with 978 µL of sodium phosphate buffer and 122 µL of MT buffer, shaken, and broken for 90 s (30 s at once) at 70 HZ on the high-throughput tissue grinder. The tube was then centrifuged for 10 min at 14,000 rcf. Finally, the FastDNA Spin Kit for Feces was used to purify the bacterial DNA in the supernatant.

The 16S rRNA gene’s V3–V4 region was then amplified by PCR using the extracted DNA as a template, in accordance with previously outlined procedures [30]. The integrity of the PCR products was assessed using Agarose gel electrophoresis, and the concentration of the PCR products was ascertained using Nanodrop (Thermo Fisher Scientific Inc, Waltham, MA, USA). For creating a library, PCR products of the same quality were extracted from each sample. A sequencing process was performed on the amplified products using an Illumina MiSeq (San Diego, CA, USA). DADA2 was used to file the raw data, which were then examined using Qiime 2 [31]. 

### 2.7. Statistical Analyses

The mean ± standard error of the mean (SEM) was used to show the results. When the data had a normal distribution, one-way analysis of variance (ANOVA) was utilized to compare the differences between groups of more than two. The post hoc Tukey’s test was employed to evaluate statistically significant variations across groups. When the data were not normally distributed, the Kruskal–Wallis test was used to compare the medians between the groups [32]. The male and female mice were contrasted with their respective control groups. The alpha diversity of male and female mice was evaluated using Qiime2, which included the Chao1 and Shannon indexes, relative to their respective control groups. By using the R (version 4.3.2) packages “vagan”, “ape”, and “ggplot2”, PCoA (Principal Co-ordinates Analysis), based on the distance between the Bray–Curtis matrix, was used to compute beta diversity [32]. The Qiime 2 was used for data and bioinformation analysis. The sequences excluded the expected, errors and samples were screened for chimeras. The accession number for sequencing was PRJNA884502. Also, *, *p* < 0.05, **, *p* < 0.01, and ***, *p* < 0.001, compared to the control group in every figure. The mean ± SEM (*n* = 8 per group) represented the data.

## 3. Results

### 3.1. B. longum Subsp. infantis Influenced IgA and sIgA Levels in the Colon of Mice

Strains I4MI, I10TI, and B6MNI increased the level of IgA in both female and male mice (*p* < 0.05, Figure 2A), whereas only B6MNI increased the level of sIgA significantly in female mice (*p* < 0.01, Figure 2B). Additionally, I4MI, I4MNI, I8TI, I10TI, and B6MNI increased the level of IgA (*p* < 0.05, Figure 2C) significantly, but no significance was found for the increase of the level of sIgA in male mice (Figure 2D). 

### 3.2. B. longum Subsp. infantis Influenced IgA Synthesis Genes

Based on the above results, *B. longum* subsp. *infantis* I4MI, I5TI, I10TI, and B6MNI for female mice, and I4MI, I5TI, I8TI, and B6MNI for male mice were selected for subsequent analysis. The IgA synthetic pathway in the colon was also assessed by qRT-PCR. For female mice, *B. longum* subsp. *infantis* I4MI upregulated the relative expression of BAFF, BCMA, and PIgR (*p* < 0.01, Figure 3B,D,F) significantly. I5TI increased the relative expression of BAFF, APRIL, BCMA, and PIgR (*p* < 0.01, Figure 3B–F). B6MNI increased the relative expression of BAFF, APRIL, BCMA, and PIgR (*p* < 0.05, Figure 3B–F). No significant differences were found in TGF-β and TACI expression (Figure 3A,E). For male mice, I5TI, I8TI, and B6MNI upregulated BAFF and BCMA relative expression (*p* < 0.05, Figure 3H,J). Whereas I4MI only upregulated expression of the BAFF gene (*p* < 0.05, Figure 3H). No significant differences were found in TGF-β, APRIL, TACI, and PIgR expression (Figure 3G,I,K,L).

### 3.3. B. longum Subsp. infantis Influenced IgA+ Plasmocyte Synthetic-Related Genes in PPs

PPs are an important site for an organism to produce IgA based on the stimulation of extrinsic factors. IgA+ plasmacyte synthetic-related genes were detected by qRT-PCR. In female mice, the relative expression of TGF-β and IL-21 was significantly higher in I4MI and B6MNI compared to those in control mice (*p* < 0.05, Figure 4D,H), and the relative expression of CXCR5 and CCR6 was significantly higher in I10TI and B6MNI compared to those in the control mice (*p* < 0.05, Figure 4E,G). Furthermore, I5TI increased the relative expression of CCL20 (*p* < 0.01, Figure 4J), and B6MNI increased the relative expression of PIgR (*p* < 0.05, Figure 4K). In male mice, the relative expression of BAFF, BCMA CCL9, and CCL20 was significantly higher in I4MI, I5TI, I8TI, and B6MNI compared to those in the control group (*p* < 0.05, Figure 4L,M,T,U). The TACI expression was significantly higher in I5TI, I10TI, and B6MNI groups (*p* < 0.01, Figure 4N). The TGF-β expression was significantly higher in I8TI and B6MNI groups (*p* < 0.05, Figure 4O). Moreover, I8TI and B6MNI also increased the relative expression of CXCL13 and IL-21, respectively (*p* < 0.05, Figure 4Q,S).

### 3.4. B. longum Subsp. infantis Influenced IgA Synthesis-Related Genes in Jejunum and Ileum

The small intestine, as part of the intestinal tract, plays an equally important immunological role in maintaining host health. The relative expression of IgA synthesis-related genes in the jejunum was detected. No significant difference was found in the expression of IgA synthesis-related genes in the jejunum of female mice (Figure 5A–F). For male mice, I5TI increased the relative expression of TACI mRNA (*p* < 0.05, Figure 5K).

*B. longum* subsp. *infantis* influenced IgA synthesis-related genes in the ileum of mice. For IgA+ synthesis-related genes, I4MI and I5TI increased the relative expression of BAFF (*p* < 0.05, Figure 6B) and TACI in females, respectively (*p* < 0.05, Figure 6E). Additionally, I4MI, I5TI, and B6MNI upregulated of BAFF expression in males (*p* < 0.05, Figure 6H). 

### 3.5. Variety of sIgA-Coated Bacterial Composition

For female mice, no significant differences were found in Chao 1 and Shannon indexes (Figure 7A,B). For PCoA, no significant differences were found for sIgA-coated bacteria among the five groups (Figure 7E). Additionally, *Escherichia-Shigella* remained the dominant sIgA-coated bacteria in the control, I4MI (32.90%), I5TI (42.72%), I10TI (60.12%), and B6MNI (19.23%) groups (Figure 8A).

For male mice, Chao 1 and Shannon indexes were significantly higher in the I4MI, I5TI, I8TI, and B6MNI groups compared with the control group (*p* < 0.05, Figure 7C,D). PCoA results for sIgA-coated bacteria showed bacterial composition among groups was different (Figure 7F). sIgA-coated *Escherichia-Shigella* was dominant in the control (85.90%), I5TI (44.18%), and B6MNI (34.7%) groups, whereas sIgA-coated *Lactobacillus* was dominant in the I4MI (26.79%) and I8TI groups (16.40%, Figure 8B). I4MI, I8TI, and B6MNI increased sIgA-coated *Alistipes* (*p* < 0.05, Figure 8C) and *Lactobacillus* (*p* < 0.05, Figure 8D) and decreased the relative abundance of *Escherichia-Shigella* (Figure 8E). I4MI, I5TI, and I8TI treatments also increased the relative abundance of unclassified Muribaculaceae (*p* < 0.05, Figure 8F).

## 4. Discussion

In this study, we assessed IgA level-related cytokines and genes, as well as alterations in the makeup of sIgA-coated bacteria, in mice aged 1–3 weeks after gavage with eight strains of *B. longum* subsp. *infantis*. According to our research, *B. longum* subsp. *infantis* can regulate colonic IgA levels in both male and female mice, but it is unable to regulate the IgA synthetic gene in the ileum and jejunum. Furthermore, these effects rely on the intestinal niches and are strain specific.

IgA, an important immunoglobulin in the intestine, plays a critical role in intestinal mucosal immunity regulation [33]. According to studies, *Bifidobacterium* can increase the production of IgA by interacting with intestinal immune cells, regulating immune response, and promoting the function of the intestinal mucosal barrier. These mechanisms all contribute to the maintenance of intestinal health and the enhancement of the intestinal immune system. Additionally, this is one of the key ways that *Bifidobacterium* contributes to preserving the balance of the gut microbiota, avoiding intestinal disorders. For instance, it has been demonstrated that *B. animalis* HY8002 raises the IgA level in PP cells [21], and *B. longum* subsp. *infantis* M-63 can also significantly increase fecal IgA levels [20]. In this study, *B. longum* subsp. *infantis* I4MI, I10TI, and B6MNI increased IgA in the colon of females. In addition, I4MI, I4MNI, I8TI, I10TI, and B6MNI increased IgA in male mice. BAFF and its receptor in the colon [33], irreplaceable at the IgA level, were activated by *B. longum* subsp. *infantis* in both males and females. *B. longum* subsp. *infantis* increases IgA secretion; however, it has a distinct effect on the genes involved in IgA synthesis. Different cell types, including dendritic cells, macrophages, and epithelial cells, create B-activating factor (BAFF), which can support B cell survival, proliferation, and differentiation and keep the body’s B cell level stable. [34]. BCMA is a receptor for BAFF, which plays an important role in the maturation and function of B cells [35]. When BAFF binds to BCMA, it can trigger a series of downstream signal transduction pathways, thereby promoting the survival and differentiation of B cells [35]. APRIL can bind to both BCMA and TACI to activate B cells, promoting IgA synthesis and secretion [36]. The results showed that *B. longum* subsp. *infantis* mainly affects the binding of BAFF and APRIL with BCMA, activates B cells to promote IgA production, and has no significant effect on TACI.

IgA synthesis occurs in intestinal PPs, which are crucial, and the process of producing IgA in plasma cells is far more intricate than it is in the intestinal lamina propria [37]. More external stimulation is needed for PPs to produce IgA plasma cells, such as IL-21, which enhances immune cells’ antigen-specific response, and CCL9 and CCL20, chemokines that draw DC into M cells [38,39]. In this study, *B. longum* subsp. *infantis* stimulated more genes related to IgA plasma cell production in PPs in male mice. The majority of these genes are found in the B cell activating factors and their receptors. This might suggest that *B. longum* subsp. *infantis* functions as an antigen in mice to deactivate B and M cells and imprint immunological memory in later life [40]. IgA synthesis-related genes were also found in the ileum and jejunum, but in this investigation, B. longum subsp. infantis only had an effect on IgA synthesis genes in the colon. This might be the case because different regions of the intestine have varied functional compositions, and the microorganism’s colonization habitats dictate how it functions [41,42]. For instance, the oligosaccharides found in human breastfeeding, which nourish *Bifidobacterium*, are difficult for the bacteria to digest and instead stay in the colon to support bacterial development. Meanwhile, *B. longum* subsp. *infantis* has an absolute advantage in the infant stomach because it has more genes for breaking down HMO than other *Bifidobacterium* species, such as *B. breve* and *B. bifidum* [29].

IgA dimers are transported from the submucosa to the mucosal surface by the secretory component (SC), an auxiliary component of the secretory IgA molecule that is produced and released by mucosal epithelial cells [43]. sIgA is formed when the secretory component binds to IgA and is released into the intestinal lumen via PIgR [44]. Loss of the mRNA of PIgR results in a lack of sIgA in the intestine [45]. It has been suggested that raising mice’s sIgA levels after weaning may have an impact on the composition and activity of the gut microbiota [46], and this effect is maintained and amplified when the mice reach adulthood. However, B6MNI was the sole treatment that enhanced IgA in the colon of female mice, while having no effect on IgA synthesis in the jejunum or ileum. This finding may be explained by strain variations and intestinal niches [47]. Furthermore, B6MNI was isolated from human breastmilk, whereas the remaining seven strains were obtained from the feces of infants. The functions of strains may also be impacted by their various origins. To find out if these eight strains of *B. longum* subsp. *infantis* vary in their genomes and metabolites in vitro and in vivo, more research is required.

The composition and relative abundance of bacteria coated with sIgA are varied with age and are supposed to be biomarkers of disease occurrence [1]. For example, infants with necrotizing enterocolitis showed a lower relative abundance of Bifidobacteriaceae [48]. sIgA-coated anaerobic bacteria significantly increase within 6–12 months [1]. This study showed the sIgA-coated bacteria were not influenced in female mice gavaged with *B. longum* subsp. *infantis*, in contrast to a study where *Lactobacillus reuteri* was gavaged in mice [49]. The age of the mice and the method of gavage differ between the results of the prior study and ours. As a result, several variables affect the changes in the composition of sIgA-coated bacteria. Nonetheless, there was a significant impact on the sIgA-coated bacteria in male mice. In the sIgA-coated version, the relative number of helpful bacteria rose, while that of opportunistic pathogenic bacteria was considerably decreased. Previous research has documented sex-dependent differences in the composition of the gut microbiota [50]. The gut microbiota of recipient mice differed according to gender when the fecal microbiota from certain pathogen-free female mice was transplanted into germ-free mice [51,52]. According to another study, females have a larger abundance of *Bifidobacterium* in the mucosa-associated microbiome [52]. Because soy iso-flavones, chemically similar to estrogen, significantly increase *Bifidobacterium*, there may be a connection between estrogen and this sex-specific variance in gut microorganisms. The varying outcomes of the probiotic intervention might have been caused by variations in the bacteria of the male and female mice. These findings suggest that an important window of immunological development is influenced by early-life microbiome colonization. For newborns at risk of microbiota imbalances that might impair immune development and increase vulnerability to illness, gender-specific microbial formulations should be developed via a knowledge of the bacteria and mechanisms involved.

Many nutrients may be obtained from a varied diet, which is advantageous for preserving the diversity of the gut microbiota and encouraging the development and regrowth of intestinal *Bifidobacterium*. Conversely, a single food structure can cause the amount of bifidobacteria in the colon to drop, which would upset the gut microbiota’s delicate equilibrium. Breast milk is mostly consumed by infants throughout their first six months of life, and the IgA it contains helps babies grow normally. Of course, the other nutrients it contains—beef milk oligosaccharides, microbes, proteins, and fats—can help support the health of a newborn. However, there are factual and subjective variables that contribute to infants using formula milk. Although the nutritional content of formula milk is close to that of breast milk, it still cannot be compared to breast milk [18]. Thus, in the era of food nutrition, supplementing with probiotics to support baby health and make up for formula milk’s shortcomings has become a worry. According to this study, the structure of the gut microbiota and IgA levels are affected differently by supplementing with different strains of *B. longum* subsp. *infantis*. This further demonstrates that the regulatory actions of probiotics vary depending on the strain. These elements will also control intestinal mucosal immunity in the early years, which promotes healthy growth in later life. Probiotic dietary supplements can fairly and successfully make up for subhealth conditions brought on by a variety of circumstances.

The application value of *B. longum* subsp. *infantis* promoting IgA production in mice in clinical aspects lies in its potential to enhance mucosal immunity and protect against infections. IgA is an important antibody and plays a crucial role in defending against pathogens at mucosal surfaces. By stimulating the production of IgA, *B. longum* subsp. *infantis* can help strengthen the immune response in the gut and respiratory tract, reducing the risk of infections. This probiotic strain may also have the potential to modulate immune responses and reduce inflammation, which could be beneficial in managing various immune-related conditions. Further research and clinical trials are needed to validate these potential clinical benefits and explore the specific applications of *B. longum* subsp. *infantis* in human health.

The limitation of this article is the lack of analysis of the characteristics of the strains themselves. Additional analysis should be performed to explain the relationship between the gene of strains and its ability to increase the level of IgA secretion. 

## 5. Conclusions

Given to mice ranging in age from one to three weeks, *B. longum* subsp. *infantis* increased colonic IgA levels in both male and female mice, as well as sIgA levels in female mice. It also regulated genes that promote IgA in the colon and in PPs. In terms of boosting IgA levels, eight strains of *B. longum* subsp. *infantis* exhibited strain-specific effects; among them, B6MNI exhibited more pronounced effects than the other strains. Furthermore, the effects of *B. longum* subsp. *infantis* on mice were intestinal niche-dependent.

## Figures and Tables

**Figure 1 nutrients-16-01148-f001:**
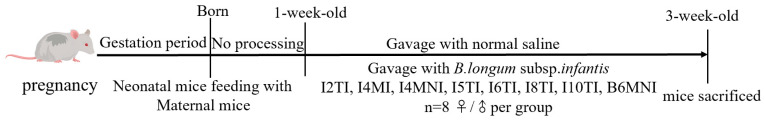
Process of animal experiment. Neonatal mice were kept with their mother until they were weaned at three weeks, after a gestation period of three weeks. From one week to three weeks old, neonatal mice were gavaged with eight strains of *B. longum* subsp. *infantis* and normal saline (female and male groups).

**Figure 2 nutrients-16-01148-f002:**
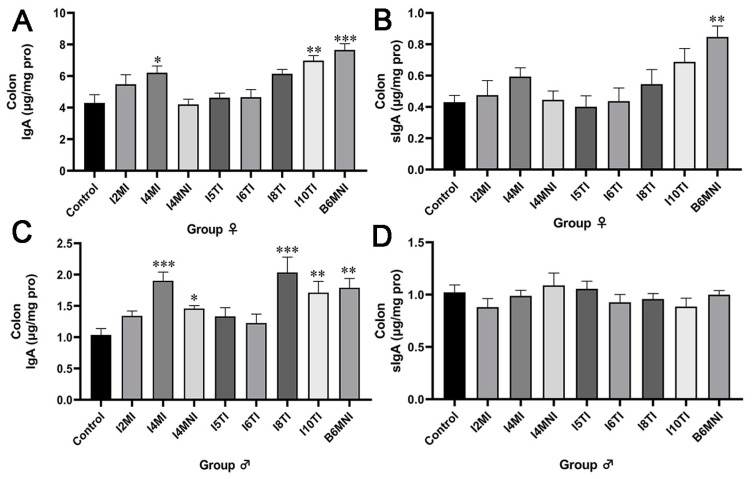
*B. longum* subsp. *infantis* influenced the level of IgA and sIgA of female (**A**,**B**) and male mice (**C**,**D**). (**A**) the level of IgA; (**B**) the level of sIgA; (**C**) the level of IgA; (**D**) the level of sIgA. *, *p* < 0.05, **, *p* < 0.01, ***, *p* < 0.001, compared to corresponding control group.

**Figure 3 nutrients-16-01148-f003:**
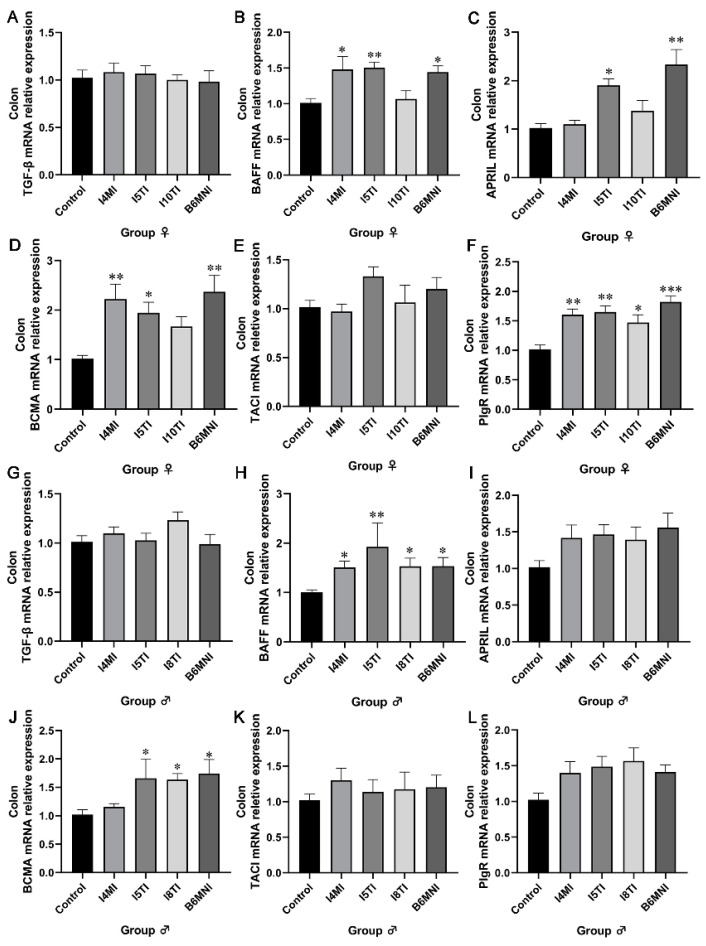
*B. longum* subsp. *infantis* influences the related genes for IgA+ plasma cell production in the colon of females (**A**–**F**) and males (**G**,**H**). (**A**) the expression of TGF-β; (**B**) the expression of BAFF; (**C**) the expression of APRIL; (**D**) the expression of BCMA; (**E**) the expression of TACI; (**F**) the expression of PIgR; (**G**) the expression of TGF-β; (**H**) the expression of BAFF; (**I**) the expression of APRIL; (**J**) the expression of BCMA; (**K**) the expression of TACI; (**L**) the expression of PIgR. * *p* < 0.05, **, *p* < 0.01, ***, *p* < 0.001, compared to corresponding control group.

**Figure 4 nutrients-16-01148-f004:**
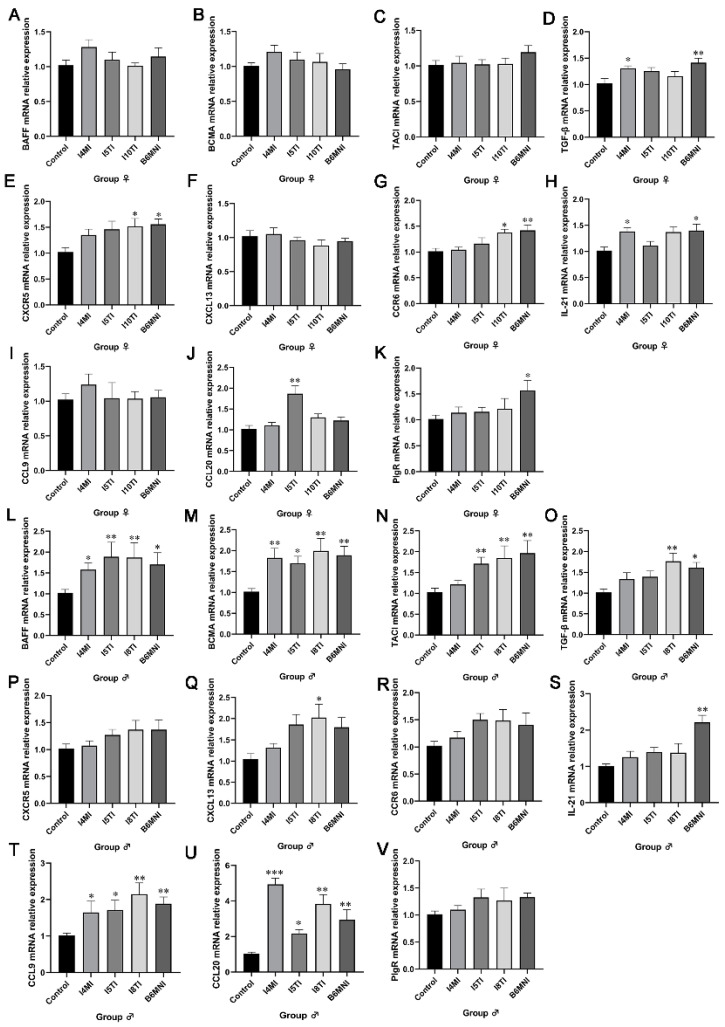
*B. longum* subsp. *infantis* influenced IgA production-related gene expression in PPs in females (**A**–**K**) and males (**L**–**V**). (**A**) BAFF; (**B**) BCMA; (**C**) TACI; (**D**) TGF-β; (**E**) CXCR5; (**F**) CXCL13; (**G**) CCR6; (**H**) IL-21; (**I**) CCL9; (**J**) CCL20; (**K**) PIgR; (**L**) BAFF; (**M**) BCMA; (**N**) TACI; (**O**) TGF-β; (**P**) CXCR5; (**Q**) CXCL13; (**R**) CCR6; (**S**) IL-21; (**T**) CCL9; (**U**) CCL20; (**V**) PIgR. * *p* < 0.05, **, *p* < 0.01, ***, *p* < 0.001, compared to corresponding control group.

**Figure 5 nutrients-16-01148-f005:**
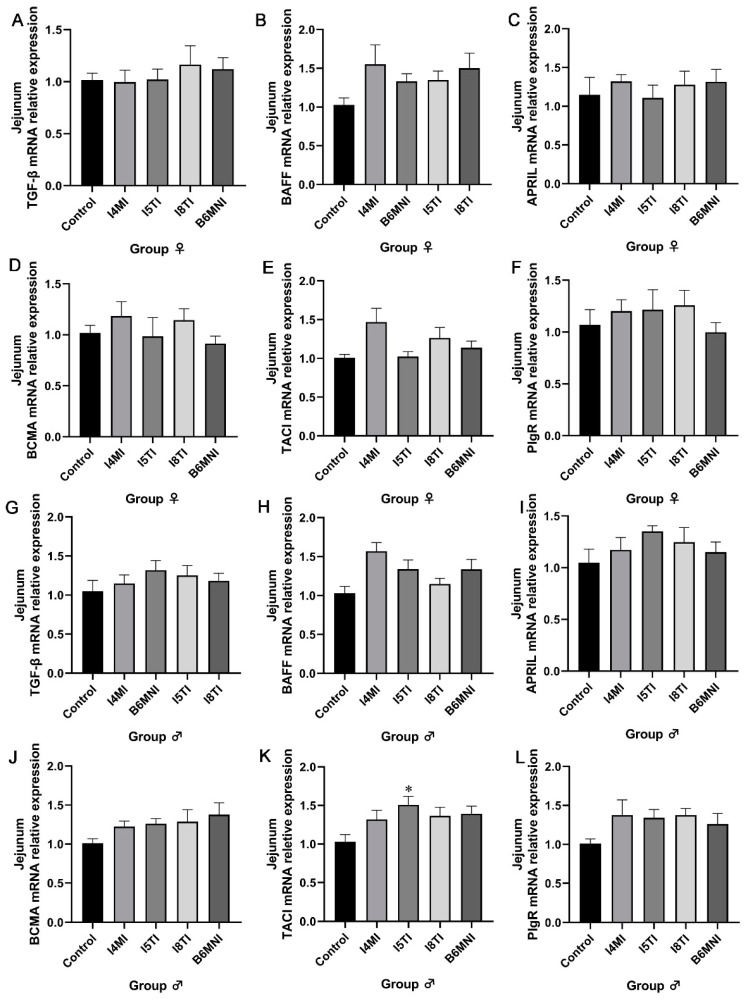
*B. longum* subsp. *infantis* influenced IgA+ plasmacyte synthesis-related gene expression in the jejunum of female (**A**–**D**) and male mice (**E**–**H**). (**A**) TGF-β; (**B**) BAFF; (**C**) APRIL; (**D**) BCMA; (**E**) TACI; (**F**) PIgR; (**G**) TGF-β; (**H**) BAFF; (**I**) APRIL; (**J**) BCMA; (**K**) TACI; (**L**) PIgR. * *p* < 0.05, compared to corresponding control group.

**Figure 6 nutrients-16-01148-f006:**
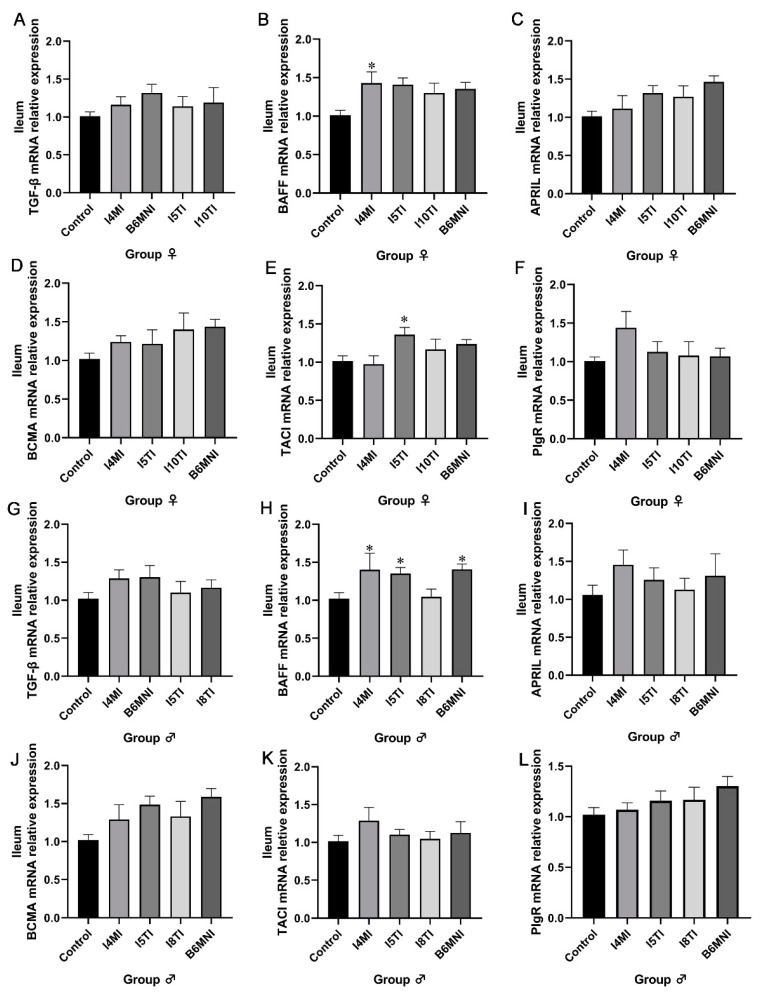
*B. longum* subsp. *infantis* influenced IgA production-related gene expression in the ileum of female (**A**–**D**) and male mice (**E**–**H**). (**A**) TGF-β; (**B**) BAFF; (**C**) APRIL; (**D**) BCMA; (**E**) TACI; (**F**) PIgR; (**G**) TGF-β; (**H**) BAFF; (**I**) APRIL; (**J**) BCMA; (**K**) TACI; (**L**) PIgR. * *p* < 0.05, compared to corresponding control group.

**Figure 7 nutrients-16-01148-f007:**
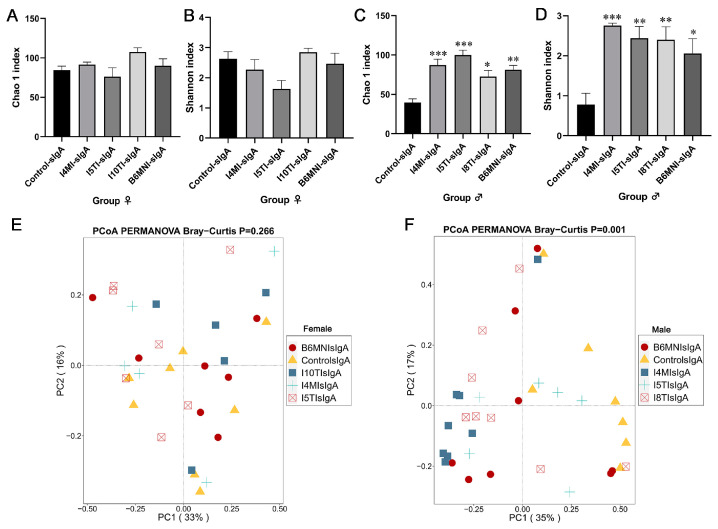
*B. longum* subsp. *infantis* influenced diversity of sIgA-coated bacteria. (**A**–**D**) α-diversity. (**E**,**F**) β-diversity. * *p* < 0.05, **, *p* < 0.01, ***, *p* < 0.001, compared to corresponding control group.

**Figure 8 nutrients-16-01148-f008:**
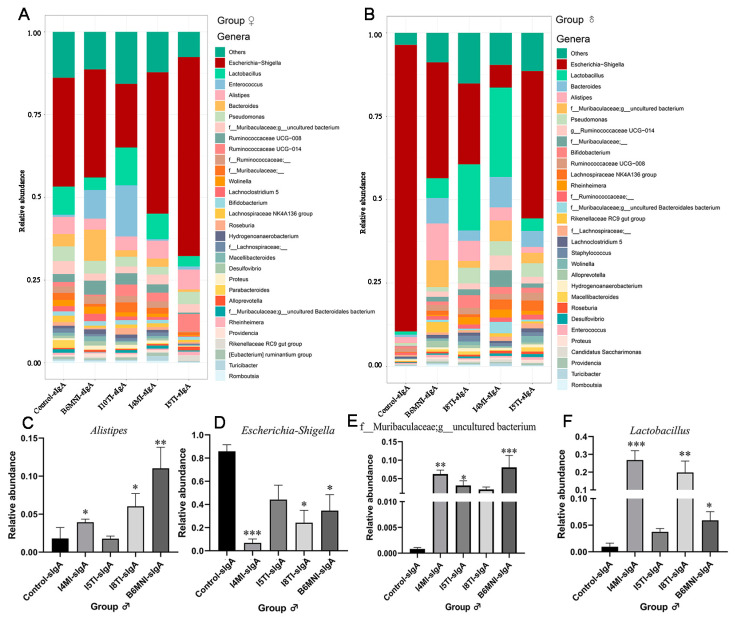
Effects of *B. longum* subsp. *infantis* on the composition of sIgA-coated bacteria. (**A**,**B**) sIgA-coated bacterial stacking plot at genus level. (**C**–**F**) Significant difference of sIgA-coated bacteria in male mice. * *p* < 0.05, **, *p* < 0.01, ***, *p* < 0.001, compared to corresponding control group.

**Table 1 nutrients-16-01148-t001:** Primers used in this study.

Primer Name	Forward (5′-3′)	Reverse (5′-3′)
TGF-β	CTCCCGTGGCTTCTAGTGC	GCCTTAGTTTGGACAGGATCTG
BAFF	GGAACAGACGCGCTTTCCA	GGCCGGTCATTACCTTTTCGT
APRIL	GGAACAGACGCGCTTTCCA	GGCCGGTCATTACCTTTTCGT
BCMA	GCGCAACAGTGTTTCCACAG	CGCTTGGATCACAGTAAGGCT
TACI	ATGGCATTCTGCCCCAAAGAT	ATGGTCGTAGTACCTGCCTTG
pIgR	ATGAGGCTCTACTTGTTCACGC	CGCCTTCTATACTACTCACCTCC
CXCR5	ATGAACTACCCACTAACCCTGG	TGTAGGGGAATCTCCGTGCT
CXCL13	GGCCACGGTATTCTGGAAGC	GGGCGTAACTTGAATCCGATCTA
CCR6	CCTGGGCAACATTATGGTGGT	CAGAACGGTAGGGTGAGGACA
IL-21	GGACCCTTGTCTGTCTGGTAG	TGTGGAGCTGATAGAAGTTCAGG
CCL9	CCCTCTCCTTCCTCATTCTTACA	AGTCTTGAAAGCCCATGTGAAA
CCL20	GCCTCTCGTACATACAGACGC	GCCTCTCGTACATACAGACGC

## Data Availability

Raw sequencing reads were accessible from the National Center for Biotechnology Information (NCBI) Sequence Read Archive (SRA) database under the accession number PRJNA884502.

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
