# Peer review of "Bifidobacterium longum Subsp. infantis Promotes IgA Level of Growing Mice in a Strain-Specific and Intestinal Niche-Dependent Manner"

_nutrients, 2024, doi:10.3390/nu16081148_

Round 1

Reviewer 1 Report

Comments and Suggestions for Authors

Dear editor and authors, in principle I appreciate the opportunity to review this work, which seems to me to be of great interest and topicality, in an area of research in which new actions of the microbiota on different processes of organisms are increasingly observed. In principle it seems to me to be a very well done work, although I think the figures should be simplified, so that both genders (males and females) could be placed in the same panel, thus better observing the action as a whole and on each of the two genders. On the other hand, a section is missing at the end of the discussion that establishes the authors' opinion on the possible actions in humans and their possible clinical applications, as well as the possible lines of research that this opens up. The latter would give added value to the work, which I repeat is carried out with great rigor.

Author Response

Comments1: Dear editor and authors, in principle I appreciate the opportunity to review this work, which seems to me to be of great interest and topicality, in an area of research in which new actions of the microbiota on different processes of organisms are increasingly observed. In principle it seems to me to be a very well done work, although I think the figures should be simplified, so that both genders (males and females) could be placed in the same panel, thus better observing the action as a whole and on each of the two genders. On the other hand, a section is missing at the end of the discussion that establishes the authors' opinion on the possible actions in humans and their possible clinical applications, as well as the possible lines of research that this opens up. The latter would give added value to the work, which I repeat is carried out with great rigor.

Response 1: Many thanks to the reviewers for their suggestions. In this article we mainly wanted to compare the effects of eight strains of B. longum subsp. infantis on intestinal and serum levels of IgA in male and female mice, respectively. So we did not put the data of male and female mice in the same panel. In that case, we need to do the significance analysis of the data related to both male and female mice at the same time. This is different from the aim of our study. Therefore, we splited the data of male and female for analysis. However, the reviewer provided a better idea for the analysis, and we will consider this in our further studies.

Furthermore, we supplied the discussions of possible clinical application of B. longum subsp. infantis in the discussions part, as follow. The application value of B. longum subsp. infantis promoting IgA production in mice in clinical aspects lies in its potential to enhance mucosal immunity and protect against infections. IgA is an important antibody that plays a crucial role in defending against pathogens at mucosal surfaces. By stimulating the production of IgA, B. longum subsp. infantis can help strengthen the immune response in the gut and respiratory tract, reducing the risk of infections. This probiotic strain may also have the potential to modulate immune responses and reduce inflammation, which could be beneficial in managing various immune-related conditions. Further research and clinical trials are needed to validate these potential clinical benefits and explore the specific applications of B. longum subsp. infantis  in human health.

Reviewer 2 Report

Comments and Suggestions for Authors

The following reference should be added to the introduction:

Mills S, Yang B, Smith GJ, Stanton C, Ross RP. Efficacy of Bifidobacterium longum alone or in multi-strain probiotic formulations during early life and beyond. Gut Microbes. 2023 Jan-Dec;15(1):2186098. doi: 10.1080/19490976.2023.2186098.

L. 100-103: You should provide more precise information regarding the origin of the strains. Were these bacterial strains mentioned in a specific publication? If so, which one? It's crucial to understand their source and characteristics, such as potential antibiotic resistance and whether their probiotic properties have been tested.

Section: 2.4. Quantitative real-time polymerase chain reaction (qRT-PCR) analysis

Could you clarify the meaning of "using the TriZol method [30]"? Reference No. 30 does not appear to mention such a method. Please provide detailed instructions on the methodology. Additionally, could you specify which Real-Time PCR system was utilized?

2.5. Separation of sIgA-coated bacteria

L 165-166: “using a magnetic rack  washed with PBS three times and stored at -80℃ for DNA extraction

The method for extracting DNA from the samples and subsequently testing them is not specified anywhere.

2.6. Analysis of 16S rRNA gene sequencing 167

MP Biomedicals (Irvine, CA, USA) manufactured the FastDNA Spin Kit for Feces,

which was used to extract DNA from fecal samples. 20 mg of feces from each mouse was  taken to extract bacterial DNA.”

The above-mentioned comment applies equally to this field.

2.7. Statistical analyses

The authors should create a small paragraph, containing the following: Data Analysis and Bioinformatics. Which platform did they use? Were sequences trimmed or excluded as expected errors? Samples were screened for chimeras? Accessions numbers for sequencing and metagenomics analyses.

Please improve the resolution of figures and increase the size of the letters in the figures.

Please discuss the methodological limitations of the research, and future aspects.

Author Response

Comments 1: Mills S, Yang B, Smith GJ, Stanton C, Ross RP. Efficacy of Bifidobacterium longum alone or in multi-strain probiotic formulations during early life and beyond. Gut Microbes. 2023 Jan-Dec;15(1):2186098. doi: 10.1080/19490976.2023.2186098.

Response 1: This reference has already added to the introduction.

Comments 2: L. 100-103: You should provide more precise information regarding the origin of the strains. Were these bacterial strains mentioned in a specific publication? If so, which one? It's crucial to understand their source and characteristics, such as potential antibiotic resistance and whether their probiotic properties have been tested.

Response 2: The strain B6MNI was isolated from human breastmilk and the other seven strains were isolated from infant feces as mentioned in the article. We haven’t determined the characteristics of these strains but it’s on the way. We appreciated reviewer’s suggestions and will focus on the characteristics of these strains deeply in the future.

Comments 3: Section: 2.4. Quantitative real-time polymerase chain reaction (qRT-PCR) analysis. Could you clarify the meaning of "using the TriZol method [30]"? Reference No. 30 does not appear to mention such a method. Please provide detailed instructions on the methodology. Additionally, could you specify which Real-Time PCR system was utilized?

Response 3: The TriZol method and Real-Time PCR system have already supplemented in the section 2.4.

Comments 4: 2.5. Separation of sIgA-coated bacteria

L 165-166: “using a magnetic rack washed with PBS three times and stored at -80℃ for DNA extraction

The method for extracting DNA from the samples and subsequently testing them is not specified anywhere.

Response 4: The method for extracting DNA was in the section 2.6 and we have corrected the title of section 2.6. The DNA of sIgA-coated bacteria was extracted through MP Biomedicals (Irvine, CA, USA) manufactured the FastDNA Spin Kit. And the specific method has been supplemented as follow.

The sIgA-coated bacteria isolated in section 2.5 was taken to extract bacterial DNA. Briefly, a Lysing Matrix E tube was filled with bacterial sediment. After adding and vortexing for 10 to 15 seconds, 825 µL of sodium phosphate buffer and 275 µL of pre-lysis solution dissolving solution were added. Centrifugation at 14,000 rcf for 5 minutes was then performed, and the supernatant was disposed of. The mixture was then mixed with 978 µL of sodium phosphate buffer and 122 µL of MT buffer, shaken, and broken for 30 s (3~5 times) at 70 HZ on the high-throughput tissue grinder. The Lysing Matrix E tube was then centrifuged for 10 minutes at 14,000 rcf. At last, the FastDNA Spin Kit for Feces was used to purify the bacterial DNA in the supernatant.

Comments 5: 2.6. Analysis of 16S rRNA gene sequencing 167

MP Biomedicals (Irvine, CA, USA) manufactured the FastDNA Spin Kit for Feces,

which was used to extract DNA from fecal samples. 20 mg of feces from each mouse was taken to extract bacterial DNA.”

The above-mentioned comment applies equally to this field.

Response 5: We rewritten this section and supplied the DNA extraction method as follow.

The sIgA-coated bacteria isolated in section 2.5 was taken to extract bacterial DNA. Briefly, a Lysing Matrix E tube was filled with bacterial sediment. After adding and vortexing for 10 to 15 seconds, 825 µL of sodium phosphate buffer and 275 µL of pre-lysis solution dissolving solution were added. Centrifugation at 14,000 rcf for 5 minutes was then performed, and the supernatant was disposed of. The mixture was then mixed with 978 µL of sodium phosphate buffer and 122 µL of MT buffer, shaken, and broken for 30 s (3~5 times) at 70 HZ on the high-throughput tissue grinder. The Lysing Matrix E tube was then centrifuged for 10 minutes at 14,000 rcf. At last, the FastDNA Spin Kit for Feces was used to purify the bacterial DNA in the supernatant.

Comments 6: 2.7. Statistical analyses

The authors should create a small paragraph, containing the following: Data Analysis and Bioinformatics. Which platform did they use? Were sequences trimmed or excluded as expected errors? Samples were screened for chimeras? Accessions numbers for sequencing and metagenomics analyses.

Response 6: We have supplied the information in section 2.7 as follow.

The Qiime 2 was used for data and bioinformation analysis. The sequences were ex-cluded the expected errors and samples were screened for chimeras. Accessions num-bers for sequencing was PRJNA884502.

Comments 7: Please improve the resolution of figures and increase the size of the letters in the figures.

Response 7: The figure has already improved.

Comments 8: Please discuss the methodological limitations of the research, and future aspects.

Response 8: We have supplied the limitation and future aspects in the discussion section, as follow.

The limitation of this article is the lack of analysis of the characteristics of strains itself. More analysis should be done to explain the relationship between gene of strains and its ability to increase the level of IgA secretion.